# Household transmission of the SARS-CoV-2 Omicron variant in Denmark

Frederik Plesner Lyngse [1,2,3] ✉, Laust Hvas Mortensen[4,5], Matthew J. Denwood [6], Lasse Engbo Christiansen [7], Camilla Holten Møller[3], Robert Leo Skov [3], Katja Spiess [3], Anders Fomsgaard [3], Ria Lassaunière [3], Morten Rasmussen[3], Marc Stegger [8], Claus Nielsen[3], Raphael Niklaus Sieber [8], Arieh Sierra Cohen[3], Frederik Trier Møller [3], Maria Overvad[3], Kåre Mølbak [3,6], Tyra Grove Krause[3] & Carsten Thure Kirkeby[6]

In late 2021, the Omicron SARS-CoV-2 variant overtook the previously dominant Delta variant, but the extent to which this transition was driven by immune evasion or a change in the inherent transmissibility is currently unclear. We estimate SARS-CoV-2 transmission within Danish households during December 2021. Among 26,675 households (8,568 with the Omicron VOC), we identified 14,140 secondary infections within a 1–7-day follow-up period. The secondary attack rate was 29% and 21% in households infected with Omicron and Delta, respectively. For Omicron, the odds of infection were 1.10 (95%-CI: 1.00-1.21) times higher for unvaccinated, 2.38 (95%-CI: 2.23-2.54) times higher for fully vaccinated and 3.20 (95%-CI: 2.67-3.83) times higher for booster-vaccinated contacts compared to Delta. We conclude that the transition from Delta to Omicron VOC was primarily driven by immune evasiveness and to a lesser extent an inherent increase in the basic transmissibility of the Omicron variant.

The SARS-CoV-2 variant B.1.1.529, which is referred to as the Omicron variant of concern (VOC), overtook the Delta VOC as the most prevalent strain in South Africa during late 2021 and has since spread rapidly to at least 28 countries in Europe[1], Asia, the Middle East and South America[2,3]. Early estimates reported the Omicron VOC to be three to six times as infectious as previous variants[4], with a short doubling time[5], including early estimates from countries with a high vaccination coverage indicating doubling times of 1.6 days (Denmark), 1.8 days (UK), 2.0 days (United States), and 2.4 days (Scotland)[6]. Transmission of the Omicron VOC has been high among individuals that are vaccinated against SARS-CoV-2

infection as well as among individuals with a history of SARS-CoV-2 infection[7].

The apparent ability of the Omicron VOC to evade immunity induced by the currently used vaccines is of substantial concern worldwide: a preliminary meta-analysis of neutralization studies indicated that the vaccine effectiveness is reduced to around 40% against symptoms and to 80% against severe disease, but that the effect for booster vaccinations is at 86% and 98%, respectively[8]. These results are supported by a study indicating that the effectiveness of the Pfizer-BioNTech vaccine against infection is only 35% for the Omicron VOC[9]. This was corroborated by another in vitro study reporting an 8.4-fold

[1]Department of Economics & Center for Economic Behavior and Inequality, University of Copenhagen, Øster Farimagsgade 5, DK-1353 Copenhagen K, Denmark. [2]Danish Ministry of Health, Holbergsgade 6, DK-1057 Copenhagen K, Denmark. [3]Statens Serum Institut, Artillerivej 5, DK-2300 Copenhagen S, Denmark. [4]Statistics Denmark, Sejrøgade 11, DK-2100 Copenhagen, Denmark. [5]Department of Public Health, Faculty of Health and Medical Sciences, University of Copenhagen, Øster Farimagsgade 5, DK-1353 Copenhagen K, Denmark. [6]Department of Veterinary and Animal Sciences, Faculty of Health and Medical Sciences, University of Copenhagen, Grønnegårdsvej 8, DK-1870 Frederiksberg C Copenhagen, Denmark. [7]Department of Applied Mathematics and Computer Science, Dynamical Systems, Technical University of Denmark, Richard Petersens Plads, 324, DK-2800 Kgs. Lyngby, Denmark. [8]Department of Bacteria, Parasites and Fungi, Statens Serum Institut, Artillerivej 5, DK-2300 Copenhagen S, Denmark. ✉e-mail: fpl@econ.ku.dk

reduction in neutralization for the Omicron VOC vs. the PV-D614G reference strain, whereas there was only a 1.6-fold reduction in neutralization for the Delta VOC[10]. Therefore, the advantage of the Omicron VOC seems to be a combination of high transmissibility and increased immune-evading ability. Studies on the transmission of the Omicron VOC are few, which is a substantial gap in our knowledge of this variant worldwide[11]. In particular, it is important to clarify the extent to which the competitive advantage of the Omicron VOC can be ascribed to immune evasiveness, i.e., a higher proportion of vaccinated or previously infected individuals being susceptible to infection, an increased inherent transmissibility for this variant, or both.

The aim of this study is to investigate household transmission associated with the Omicron VOC. Specifically, we address the following questions: (1) Is the secondary attack rate higher for the Omicron VOC than for the Delta VOC? (2) Does the Omicron VOC show a higher immune evasiveness relative to the Delta VOC? (3) Is booster vaccination effective for reducing transmission?

## Results

A total of 8,568 primary cases with the Omicron VOC and 18,107 primary cases with the Delta VOC were included (Table 1). A larger proportion of primary cases with the Omicron VOC were aged 20–30 years and resided in households with 2 members than what was observed for the Delta VOC. Overall, the SAR was 29% in households with the Omicron VOC and 21% in households with the Delta VOC. The estimated SAR was also generally higher in households infected with the

Omicron VOC than for those infected with the Delta VOC across all household contact categories. Unvaccinated contacts experienced similar attack rates in households with the Omicron VOC compared to the Delta VOC (28% and 27%, respectively), while fully vaccinated individuals experienced secondary attack rates of 30% in households with the Omicron VOC and 19% in households with the Delta VOC. For booster-vaccinated individuals, Omicron was associated with a SAR of 23%, while the corresponding estimate for Delta was only 11%. See Appendix Section 3 for further summary statistics, including SARs stratified by the primary case level and more details on the "Fully vaccinated" category.

We found that the cumulative probability of contacts being tested at least once increased from 36% to 88% for Omicron contacts (blue) and from 41% to 89% for Delta contacts (red) at 7 days after the primary case tested positive (Fig. 1a). The probability of contacts being tested twice increased from 9% to 73% for Delta contacts and from 10% to 70% for Omicron contacts 7 days after the primary case tested positive. The test probability was slightly higher when the primary case was infected with the Delta VOC compared to the Omicron VOC. The probability of contacts testing positive increased from 3% and 5% on day 1, to 21% and 29% on day 7, when the primary case was infected with the Delta VOC and Omicron VOC, respectively (Fig. 1b).

The effect of vaccination on susceptibility and infectiousness of SARS-CoV-2 within households is shown in Table 2. The estimates of susceptibility by vaccine status were stratified by variant because we observed an interaction between variant and vaccination status of the

## Table 1 | Summary Statistics (primary cases and contacts reported separately)

| | Omicron | | | | Delta | | | |
|---|---|---|---|---|---|---|---|---|
| | Primary Cases | Household Contacts | Secondary Cases | SAR (%) | Primary Cases | Household Contacts | Secondary Cases | SAR (%) |
| **Total** | 8568 | 18,038 | 5229 | 29 | 18,107 | 42,964 | 8911 | 21 |
| **Sex** | | | | | | | | |
| Male | 4417 | 8714 | 2378 | 27 | 9257 | 21,126 | 4213 | 20 |
| Female | 4151 | 9324 | 2851 | 31 | 8850 | 21,838 | 4698 | 22 |
| **Age** | | | | | | | | |
| 0-10 years | 417 | 2704 | 716 | 26 | 4475 | 8550 | 2004 | 23 |
| 10-20 years | 1875 | 3506 | 834 | 24 | 3507 | 8259 | 1195 | 14 |
| 20-30 years | 2755 | 3712 | 969 | 26 | 2432 | 3905 | 645 | 17 |
| 30-40 years | 1186 | 1885 | 734 | 39 | 1909 | 7109 | 1671 | 24 |
| 40-50 years | 1094 | 3097 | 993 | 32 | 2312 | 8926 | 1858 | 21 |
| 50-60 years | 874 | 2366 | 757 | 32 | 2056 | 4099 | 960 | 23 |
| 60-70 years | 280 | 545 | 170 | 31 | 1019 | 1439 | 440 | 31 |
| 70+ years | 87 | 223 | 56 | 25 | 397 | 677 | 138 | 20 |
| **Household size** | | | | | | | | |
| 2 persons | 3339 | 3339 | 1266 | 38 | 5564 | 5564 | 1584 | 28 |
| 3 persons | 2102 | 4204 | 1179 | 28 | 3863 | 7726 | 1552 | 20 |
| 4 persons | 2190 | 6570 | 1894 | 29 | 5632 | 16,896 | 3451 | 20 |
| 5 persons | 760 | 3040 | 734 | 24 | 2462 | 9848 | 1884 | 19 |
| 6 persons | 177 | 885 | 156 | 18 | 586 | 2930 | 440 | 15 |
| **Vaccination status** | | | | | | | | |
| Unvaccinated[a] | 1166 | 4171 | 1155 | 28 | 8611 | 13,750 | 3718 | 27 |
| Fully vaccinated[b] | 6934 | 12,555 | 3768 | 30 | 8968 | 26,341 | 4875 | 19 |
| Booster vaccinated | 468 | 1312 | 306 | 23 | 528 | 2873 | 318 | 11 |

The secondary attack rate (SAR) is expressed as a percentage (%). Summary statistics based on primary cases are shown separately from summary statistics on household contacts, secondary cases and SAR. For example, there were 417 primary cases aged 0–10 years with Omicron and a total of 2704 contact aged 0–10 years living in households infected with the Omicron VOC. Of the 2,704 household contacts, 716 tested positive, yielding a SAR of 26%. Thus the SAR reflects the proportion of household contacts that tested positive, irrespective of the characteristics of the primary case.
[a]Unvaccinated includes individuals with partial vaccination.
[b]Fully vaccinated includes unvaccinated individuals with the previous infection. See Appendix Section 2 for additional summary statistics of primary cases and contacts, including more details on the "Fully vaccinated" category.

**a** Probability of testing

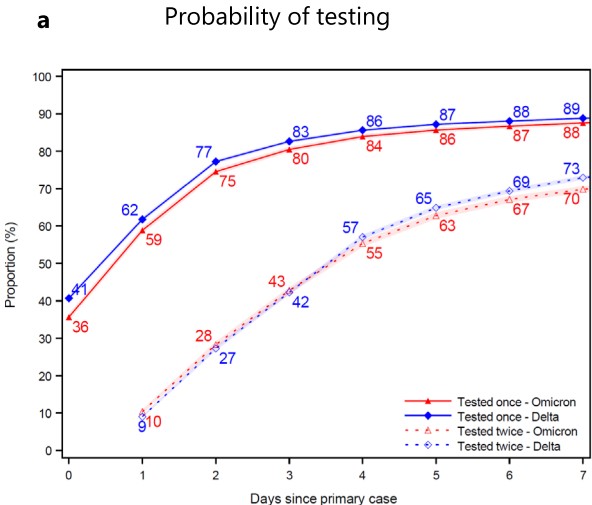

**b** Probability of testing positive

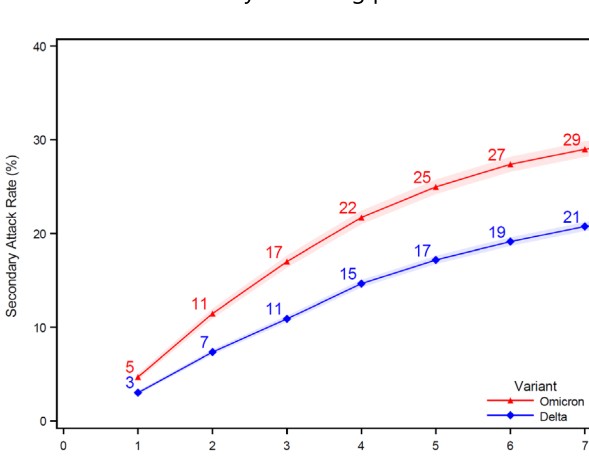

**Fig. 1 | Probability of being tested and testing positive.** Panel (**a**) shows the probability of household contacts being tested after a primary case has been identified within the household. Panel (**b**) shows the probability of contacts that test positive subsequently to a primary case being identified within the household. Note that the latter is not conditional on being tested, i.e., the denominator contains test negative individuals and untested individuals. The *x* axes show the days since the primary case tested positive, and the *y* axes show the proportion of individuals either being tested (**a**) or testing positive (**b**) with either antigen or RT-

PCR tests, based on the variant of the primary case. The SAR for each day relative to the primary case can be read directly from **b**. For example, the SAR on day 7 is 29% for Omicron (red) and 21% for Delta (blue), whereas the SAR on day 4 is 22% for Omicron and 15% for Delta. The markers show the point estimates of the mean. The shaded areas show the 95% confidence bands clustered on the household level. See Appendix Fig. S7 for the same two panels, only using RT-PCR tests, and Appendix Fig. S8 for a 14-day follow-up.

contacts ($p < 0.001$). No interaction between variant and vaccination status of the primary case was observed ($p = 0.14$).

For households infected with the Delta VOC, we estimated an OR of infection of 2.36 (95% confidence interval, CI: 2.20–2.54) for unvaccinated contacts compared to fully vaccinated contacts, and an OR of 0.41 (CI: 0.36–0.47) for booster-vaccinated contacts compared to fully vaccinated contacts, after adjustment for confounders (age and sex of the primary case, age and sex of the contact, and household size). The corresponding OR estimates for households infected with the Omicron VOC was 1.09 (CI: 0.99–1.20) for unvaccinated contacts and 0.55 (CI: 0.48–0.63) for booster-vaccinated contacts, both

compared to fully vaccinated contacts. With no interaction between vaccine status and variant, unvaccinated primary cases were associated with an OR of infection of 1.37 (CI: 1.27–1.47) compared to fully vaccinated primary cases, while booster-vaccinated primary cases were associated with a decreased OR of infection of 0.80 (CI: 0.69–0.92). This demonstrates a baseline association between vaccination status and both susceptibility and infectiousness.

The relative difference in SAR between the Omicron and Delta variants when comparing contacts with the same vaccination status is shown in Table 3. We estimated an OR of 1.10 (CI: 1.00–1.21) when comparing unvaccinated contacts living in households infected with the Omicron VOC relative to unvaccinated contacts living in households infected with the Delta VOC. Similarly, we found an OR of 2.38 (CI: 2.23–2.54) when comparing fully vaccinated contacts between variants, and an OR of 3.20 (CI: 2.67–3.83) when comparing booster-vaccinated contacts between variants.

We found an overall increased susceptibility with age of the household contact. Furthermore, we found an increased infectiousness with increasing age of adult primary cases and a decreased infectiousness with increasing age for children, suggesting a J-shape of age and infectiousness (Appendix Table S18).

We compared the Ct values of primary case samples with the Omicron VOC and the Delta VOC (Appendix Fig. S3). The distribution of Ct values for primary cases with the Omicron VOC were slightly skewed to the left compared to cases with the Delta VOC, but the median values (27.3 and 28.2, respectively) did not differ substantially. Adjustment for Ct values of the primary cases did not materially alter the findings, suggesting that the increased transmission of the Omicron VOC cannot be explained by differences in the viral load of the primary cases (Appendix Table S21). Similarly, the distribution of time since last vaccination/booster/infection among positive secondary cases were similar across the two variants (Appendix Fig. S6). However, these analyses are limited by the fact that vaccine roll out in Denmark is largely determined by age.

The probability that a sample was selected for Variant PCR was stable across Ct value and age, varying between 97% and 99% (Appendix Fig. S1). The Variant PCR test for identifying the Omicron

### Table 2 | Effect of Vaccination

|  | Susceptibility (Household contacts) | | Infectiousness (Primary case) |
|---|---|---|---|
|  | Omicron households | Delta households | All households |
| Unvaccinated[a] | 1.09 (0.99–1.20) | 2.36 (2.20–2.54) | 1.37 (1.27–1.47) |
| Fully vaccinated[b] | Ref (.) | Ref (.) | Ref (.) |
| Booster vaccinated | 0.55 (0.48–0.63) | 0.41 (0.36–0.47) | 0.80 (0.69–0.92) |

This table shows odds ratio estimates for susceptibility and infectiousness by vaccination status. Number of observations = 61,002; Number of households = 26,675. Column 1 shows the susceptibility based on the vaccination status of the household contact, conditional on being in a household infected with the Omicron VOC. Column 2 shows the susceptibility based on the vaccination status of the contact, conditional on being in a household infected with the Delta VOC. Column 3 shows the infectiousness based on the vaccination status of the primary case, unconditional on the variant in the household. Note, all estimates are from the same model, but with a different reference category across column 1–3. The estimates were adjusted for age and sex of the primary case, age and sex of the contact, and household size. The estimates are furthermore adjusted for vaccine status of the contact interacted with the household variant, and the vaccine status of the primary case. 95%-confidence intervals are shown in parentheses with cluster-robust standard errors at the household level. The odds ratio estimates for the full model are presented in Appendix Table S18, column I.
[a]Unvaccinated includes individuals with partial vaccination.
[b]Fully vaccinated includes unvaccinated individuals with previous infection.

**Table 3 | Effect of the Omicron VOC relative to the Delta VOC**

| | Unvaccinated (Contact) | Fully vaccinated (Contact) | Booster vaccinated (Contact) |
|---|---|---|---|
| Omicron households | 1.10 (1.00–1.21) | 2.38 (2.23–2.54) | 3.20 (2.67–3.83) |
| Delta households | Ref (.) | Ref (.) | Ref (.) |

This table shows odds ratio estimates for the effect of living in a household infected with the Omicron VOC relative to the Delta VOC when comparing contacts with the same vaccination status. Number of observations = 61,002; Number of households = 26,675. Column 1 shows the relative transmission of the Omicron VOC, conditional on being unvaccinated. Column 2 shows the relative transmission of the Omicron VOC, conditional on being fully vaccinated. Column 3 shows the relative transmission of the Omicron VOC, conditional on being booster vaccinated. Note, all estimates are from the same model, but with a different reference category across column 1-3. The estimates are adjusted for age and sex of the primary case, age and sex of the contact, and household size. The estimates are furthermore adjusted for vaccine status of the contact interacted with the household variant, and the vaccine status of the primary case. 95%-confidence intervals are shown in parentheses with cluster-robust standard errors clustered on the household level. The odds ratio estimates for the full model are presented in Appendix Table S18, column I.

VOC was validated with high-quality whole genome sequencing (WGS) data, which showed that 0.33% of the cases identified by the Variant PCR test as Omicron were in fact not Omicron and 0.88% of the cases identified by the Variant PCR as not Omicron were in fact Omicron (Appendix Table S5). This demonstrates the high accuracy of the Variant PCR test.

We found limited evidence of misclassification of primary and secondary cases distorting our results (Appendix Section 4.2). (i) We found no evidence of a differential effect of tertiary cases being misclassified as secondary cases, when comparing the relative SAR of two-person and multi-person households across variants (Appendix Table S13). (ii) Our results were robust to only including households where all contacts had tested negative after the primary case, thereby eliminating potential misclassification of primary cases (Appendix Table S21). (iii) We found that 98% of all secondary cases were infected with the same variant as the primary case (Appendix Table S15). We estimated this probability for households located in municipalities that had a high overall incidence of the other variant to investigate if community transmission was a major factor. We found the highest misclassification was due to community infection for households infected with the Delta VOC that were located in municipalities with a high Omicron incidence, where we found that 94% of secondary cases were infected with the same variant as the primary case (Appendix Table S16). Lastly, we conducted a sensitivity analysis of the effect of misclassifying outside-of-household infections as secondary cases with 10% and 30% misclassification. This analysis shows that our estimates are potentially biased upwards, but also that the maximum possible impact of these effects on our results are small (Appendix Table S17).

## Discussion

Our results show that the Omicron VOC is generally 2.4–3.2 times more transmissible than the Delta VOC among vaccinated household contacts, but similar in transmissibility among unvaccinated household contacts. This observation is in line with data from Public Health England[12], which estimated that 19% of Omicron VOC primary cases in households in the UK resulted in at least one other infection within the household, compared to only 8.3% of those associated with the Delta VOC. Furthermore, we show that fully vaccinated and booster-vaccinated contacts are generally less susceptible to infection compared to unvaccinated contacts (Table 2).

Overall, the findings indicate that the increased transmission of the Omicron VOC can be ascribed to immune evasion rather than an inherent increase in the basic infectiousness. If this observation can be confirmed by independent studies, it has important ramifications for the understanding of the current challenges for control of the epidemic. Bartha et al.[13] note that the existing circulating immunity within a country is of major importance in limiting the severity of the epidemic with the Omicron VOC. Although we showed that booster vaccination did offer some protection against household transmission, the reduced level of protection means that vaccination with the current vaccines is less likely to be sufficient to curb transmission with Omicron compared to previous variants. Furthermore, the duration of the protective effect is currently unknown, and the rapidly waning effectiveness of the second dose against the Omicron VOC as well as data from neutralization assays[14, 15] do raise some concerns about the longevity of the booster response. This means that the current vaccines are unlikely to mitigate the long-term spread of the Omicron VOC to the extent that has been achieved for previous variants. We therefore suggest that adapted or improved vaccines may be necessary to mitigate the spread of the Omicron VOC. However, both a primary series and a booster dose is likely to play an important role in reducing transmission on a short term and modifying the outcome of infection by reducing severity. Our estimates are important for decision makers worldwide, as they may be used to inform prediction models and thus be utilized for balancing the level of restrictions to control transmission in different situations.

The SAR was found to be higher for the Omicron VOC than for the Delta VOC across all age groups (Table 1 and Appendix Table S6). Furthermore, we found an increased susceptibility with age and an increased infectiousness with age for adults and a decreased infectiousness by age for children, suggesting a J-shape of age and infectiousness. These patterns have also been found in previous studies with previous variants and prior to vaccination roll-out[16, 17], implying that it is not a result of the age-related prioritization in the Danish vaccination strategy. The increased infectiousness for younger children is likely a result of the higher need for care for younger children compared to older children. These transmission patterns of age have implications for, e.g., care facilities, highlighting the need for increased protection against transmission now that the Omicron VOC has become dominant in many countries.

There are some potential biases in this study. Firstly, the initial spread of the Omicron VOC was characterized by spreading events, so had therefore not yet propagated evenly throughout the population by the end of our study period[3]. This could potentially impact the comparability of our estimates for the two the variants. However, our results are robust to excluding households with primary cases <10 years and including only 2-person households (Appendix Tables S19, S20).

Our analysis strongly assumes that the timing of positive tests within households can be used to infer primary and secondary infections within households. It is likely that this study misclassifies a small proportion of secondary cases, but our sensitivity analyses suggest that the potential bias in terms of the comparison between the Delta and Omicron VOC is limited (Appendix Section 4.2). By the end of 2021, self-testing kits had become widely available for purchase in Denmark. This could influence the results, for instance if individuals that self test at home refrain from also being tested in public-testing facilities, meaning that their test results are not registered in the national databases. However, the probability of being tested was very high in general, and also very similar between RT-PCR and any test (Fig. 1 and Appendix Fig. S7), meaning that most positive antigen tests were confirmed with a positive RT-PCR test during this study period. We can therefore safely assume that this was also the case for positive self-tests.

We grouped individuals with a history of SARS-CoV-2 infection together with fully vaccinated individuals because we have perfect information on vaccination and only can proxy for the previous infection by a positive RT-PCR test. However, a more detailed separation of these groups does not materially change our results (Appendix Table S22).

There are also a number of other confounders that might lead to biases within our study. Vaccinations are not randomly distributed within the population, for example, because immunocompromised and other vulnerable individuals are more likely to have had access to a booster vaccine. Similarly, there are likely underlying behavioral drivers for an individual being unvaccinated. To some extent, this has been addressed by including age as an explanatory variable in the model, but the use of registry data limits our inference. However, we find little reason to suspect any bias would be differential with regard to the variant. Consequently, we believe that our findings are robust.

In conclusion, our findings suggest that the Omicron VOC has a higher SAR than the Delta VOC. While vaccination and booster vaccination does confer protection against the Omicron VOC, we conclude that the rapid spread of the Omicron VOC is likely due to immune evasiveness and to a lower extent an inherent increase in the basic transmissibility of this variant.

## Methods

### Study design and participants

The Delta VOC has been the dominant variant in Denmark since July 2021. The first Danish case infected with the Omicron VOC was detected on 22 November 2021[18], and community transmission was present by early December 2021[19]. On 8 December, Danish authorities discontinued intensive contact tracing of close contacts for cases specifically infected with the Omicron VOC. We therefore chose a study period beginning on 9 December 2021, when cases of both variants were treated approximately equally, thus reducing bias from the earlier intensified contact tracing and active case finding of the Omicron VOC[20]. The end of the inclusion period for primary cases was set to 15 December, with household contacts followed up to 7 days after the primary case, i.e., until 22 December 2021. We chose this as the end of our study period because Christmas holidays in Denmark often start on 23 December, and often include extended family visits, which interrupts the typical transmission pattern within households. For additional information on the number of new cases, proportion with Omicron and number of tests taken in Denmark during December 2021, see Appendix Section 1.

We used Danish register data for this study. All individuals in Denmark have a unique identification number, which enables cross-linking between administrative registers. Using this, we obtained person-level information on all residential addresses from the central person registry, complete data on all antigen and RT-PCR tests for SARS-CoV-2 from the Danish Microbiology Database (MiBa[21]), and all vaccination records from the Danish Vaccination Register[22].

We identified households in Denmark using their unique residential address and assigned the same household identifier to all individuals registered at that address: this was used to define the household size. We included only households with 2–6 members to exclude care facilities and other places, where many individuals share the same address.

We defined a primary case as the first individual within a household to test positive with an RT-PCR test within the study period. We followed all tests of other household members in the study period. A secondary case was defined by either a positive RT-PCR test or a positive antigen test[23]. Almost all samples that tested positive with RT-PCR were subsequently tested with Variant PCR to determine the VOC[24] (Appendix Table S1 and Fig. S1). Based on the Variant PCR test result of the primary case, we classified households as being associated with either the Omicron or the Delta VOC. The Delta VOC has been the dominant variant in Denmark since early July 2021, accounting for close to all positive RT-PCR samples August-November 2021[25]. We excluded households with a positive RT-PCR test 60 days prior to the primary case and households where the primary case was ambiguous because two individuals tested positive on the same day.

We classified individuals into three groups: (i) unvaccinated; (ii) fully vaccinated; or (iii) booster vaccinated. The definition of fully vaccinated included individuals that had been infected more than 14 days previously, but was otherwise defined according to the vaccine used as follows: 7 days after second dose of Comirnaty (Pfizer/BioNTech); 15 days after second dose of Vaxzevria (AstraZeneca); 14 days after second dose of Spikevax (Moderna); 14 days after vaccination with Janssen (Johnson & Johnson); 14 days after the second dose for cross vaccinated. Booster-vaccinated was defined as 7 days following the booster vaccination[26, 27]. As of 22 December 2021, the distribution of vaccines in Denmark was: 85% Comirnaty, 14% Spikevax, 1% Janssen, and approximately 0% AstraZeneca[28]. All other individuals, including 59 partially vaccinated individuals, were regarded as unvaccinated.

### Statistical analyses

The causal effect of household exposure to the Omicron VOC rather than the Delta VOC on the SAR may be confounded. This is evident from the difference in characteristics between households exposed to the Omicron and the Delta VOC, the latter of which was more widely dispersed at the beginning of the study period (Table 1 and Appendix Fig. S1b). We assume that these differences are caused by the time-space patterns of transmission of the Omicron VOC when first introduced. A causal interpretation of our findings is conditional on the assumption that all effects of the non-random assignment of variants to households is intercepted by the observed household characteristics. The causal assumptions are described in Appendix Section 2.

We defined the secondary attack rate (SAR) as the within-household proportion of household contacts that were defined as secondary cases[16]. Adjusted odds ratios (OR) of infection were estimated using multivariable logistic regression models with the binary outcome of test result of each household contact as the response variable, and the household variant (Omicron vs. Delta VOC) as the main explanatory variable of interest. Variables representing age and sex of the primary case, age and sex of the household contact, and household size (2-6 individuals) were included as additional explanatory variables to account for confounding factors. In order to test if vaccine status conferred differential protection against the Omicron and Delta VOC, we included an interaction term between vaccination status of primary cases and contacts and the variant. We found no evidence of an interaction between vaccination status of primary cases and the variant ($P = 0.14$). In particular, we estimate the following equation:

$$\log\left(\frac{\Pr(y_{c,p}=1)}{1-\Pr(y_{c,p}=1)}\right) = \text{Constant} + \text{Variant}_p + \text{VaccineStatus}_c$$
$$+ \text{Variant}_p \times \text{VaccineStatus}_c + \text{VaccineStatus}_p$$
$$+ \text{Age}_p + \text{Sex}_p + \text{HouseholdSize} + \text{Age}_c + \text{Sex}_c,$$

$$(1)$$

where $y_{c,p}$ equals one if the contact $c$ is tested positive 1–7 days after exposure to primary case $p$, and zero otherwise. $\text{Variant}_p$ determines if the primary case $p$ was infected with Omicron or Delta. VaccineStatus represents the fixed effects of vaccination status (categorical variable) for the primary case $p$ and the contact $c$. Age represents fixed effects of age in 10-year intervals (categorical variable), Sex represents fixed effects of sex, and HouseholdSize represents fixed effects of household size (categorical variable). Cluster-robust standard errors were used with clustering at the household level by using Taylor series linearization to estimate the covariance matrix of the regression coefficients[29].

We also conducted a number of supplementary analyses to support the main analysis. To test the robustness of the findings, we compared different specifications of the main logistic regression model (Appendix Section 4.4). To examine the potential mediating

role of viral load of primary cases infected with the Omicron VOC relative to the Delta VOC, we plotted the distributions of cycle threshold (Ct) values for each variant (Appendix Fig. S3). We also examined the extent to which the Ct value of the primary case could explain the difference in transmission between the variants (Appendix Table S21). Our study relies on the assumption that we correctly distinguish primary cases from secondary cases, and that household secondary cases are infected by the primary case and not from the external community. To assess this potential misclassification of cases, we performed a series of robustness checks (Appendix Section 4.2). First, to investigate the potential role of differential occurrence of tertiary cases across variants, we compared the relative SAR across two-person and multi-person households, as tertiary cases are not possible in two-person households. Second, to investigate the potential role of misclassification of primary cases, we leveraged the fact that a high proportion of contacts in our sample were tested multiple times. Because contacts without a test or secondary cases testing positive on the first test could potentially be the true primary case, we restricted our sample to only include households where all contacts had tested negative after the point at which the primary case tested positive. Third, to investigate the potential misclassification of secondary cases being infected by the outside community and not the household, we estimated the probability that secondary cases were infected with the same variant as the primary case. Additionally, to maximize the probability of identifying misclassification, we focused on households that were infected with a variant that was different to that which was most prevalent in their corresponding location.

Our study also relies on Variant PCR testing to determine if each primary case was infected with the Omicron or Delta VOC. To investigate if there was bias in the selection of samples for Variant PCR testing, we investigated the probability of sampling for Variant PCR by sample Ct value and age (Appendix Fig. S1). Furthermore, we validated the variant PCR using whole genome sequencing data (Appendix Table S5). Finally, we tested the robustness of household contacts being tested and testing positive by only using RT-PCR tests, which have higher sensitivity and specificity than antigen tests (Appendix Fig. S7).

### Ethical statement

This study was conducted using data from national registers only. According to Danish law, ethics approval is not needed for this type of research. All data management and analyses were carried out on the Danish Health Data Authority's restricted research servers with project number FSEID-00004942. The study only contains aggregated results and no personal data.

### Reporting summary

Further information on research design is available in the Nature Research Reporting Summary linked to this article.

## Data availability

The data used in this study are available under restricted access due to Danish data protection legislation. The data are available for research upon request to The Danish Health Data Authority and Statens Serum Institut and within the framework of the Danish data protection legislation and any required permission from Authorities. We performed no data collection or sequencing specifically for this study.

## Code availability

The code used for this study can be downloaded from a public repository: https://github.com/Flyngse/SARS-CoV-2_OmicronDelta_HouseholdTransmission. We used SAS 9.4 to manage and analyze the data.

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

## Acknowledgements

The authors thank Statens Serum Institut and The Danish Health Data Authority for collecting and providing access to data access. We also thank the rest of the Expert Group for Mathematical Modelling of COVID-19 at Statens Serum Institut for helpful discussions. We thank the Danish Covid-19 Genome Consortium for genotyping SARS-CoV-2 positive samples. We thank Samir Bhatt for helpful comments. We thank Carl Benjamin Simpson (Department of Economics, University of Copenhagen) for proofreading the manuscript. Frederik Plesner Lyngse is supported in part by grants from Independent Research Fund Denmark (Grant no. 9061-00035B); Novo Nordisk Foundation (grant no. NNF17OC0026542); the Danish National Research Foundation through its grant (DNRF-134) to the Center for Economic Behavior and Inequality (CEBI) at the University of Copenhagen. Laust Hvas Mortensen is supported in part by grants from the Novo Nordisk Foundation (grant no. NNF17OC0027594, NNF17OC0027812). Matthew Denwood, Lasse Christiansen, and Carsten Kirkeby receive funding from Statens Serum Institut as part of the Expert Group for Mathematical Modelling of COVID-19.

## Author contributions

F.P.L. performed all data analyses. L.E.C., M.D., F.P.L., L.H.M., and C.T.K. designed the study and devised the statistical analysis. F.P.L., C.T.K., and L.H.M. wrote the first draft. F.P.L., L.H.M., M.D., L.E.C., C.H.M., R.L.S., K.S., A.F., M.M.L., M.R., M.S., C.N., R.N.S., A.S.C., F.T.M., M.O., K.M., T.G.K., and C.T.K. contributed to the discussion, revised the first draft and approved the submitted version.

## Competing interests

The authors declare no competing interests.
