## [Peer Review File · Nature Communications]

Household transmission of the SARS-CoV-2 Omicron variant in DenmarkEditorial Note: This manuscript has been previously reviewed at another journal that is not operating a transparent peer review scheme. This document only contains reviewer comments and rebuttal letters for versions considered at *Nature Communications*.

REVIEWER COMMENTS

Reviewer #5 (Remarks to the Author):

R3 comments:

I think there are still issues with the regression equation. The left hand side should have $P[Y_{cp}=1]$, not y_{cp} (we use the observed data to provide information about the (unobserved) probability and in turn estimate the regression coefficients). The main effect of vaccine status of the contact appears to be missing from the regression equation.

R4 comments:

I think the authors have come up with some creative sensitivity analyses to address potential limitations raised by the reviewer, and I am convinced by them. My only comment would be that these are not particularly well explained in the main text. I understand that space is an issue, but the result is that the reader has to go to the Supplementary to understand these analyses, despite the attempted explanation in the text. For example: "we found a limited effect of 5% of misclassification of secondary cases being infected by the community and not the household" (line 297) – I don't understand this sentence.

Other comments:

Line 231: OR estimates for vaccine status of the primary case are conditional on primary case variant, contrary to what the authors say (variant is included in the model, therefore other ORs in the model are conditional on it). What they mean is that there was no interaction term between vaccine status and variant.

Line 287: I think a word is missing here: "0.33% of the _____ cases identified by the Variant PCR test were in fact Omicron"?

Reviewer #6 (Remarks to the Author):

The manuscript by Lyngse et al. compared the transmission dynamics between delta and omicron. The study identified a unique timing where both variants are circulating, and in a household setting where the primary and secondary cases can be more easily identified. The study provided good evidence that the rapid spread of omicron was mainly driven by immunity escape. There is a major issue on the specification of the regression model which may affect the estimates.

Major comments

1. The study design was well described.
2. Appendix 4.2 nicely considered different scenarios that contacts could potentially be primary cases (Type A & B). Could the authors provide the numbers in each category to support the classification and the likely impact of misclassification?
3. For type A & B, was it possible to utilize additional information, such as timing of the symptom onset, testing results of the contacts before confirmation of the first case in the household? This will enhance accuracy of the classification.
4. "We followed all tests of other household members in the study period." Could test results before confirmation of the primary case be used to improve classification of the primary and secondary cases?
5. The authors have carried out a series of sensitivity analyses and the results should be robust, though above could be opportunity to reduce misclassification.
6. Model equation, the interaction term $\text{Variant}_p \times \text{VaccineStatus}_c$ was included. However, it is

unclear why the main effect VaccineStatus_c was not included in the model. This is equivalent to making a strong and likely incorrect assumption that vaccination has no protective effect on the contacts within household.

7. Could the authors comment on whether Ct values for omicron vs delta have the same interpretation?

Minor comments

8. Introduction, "The Omicron VOC has been reported to be three to six times as infectious as previous variants". Reference 4 was based on very early estimates in late 2021 and 3-6 times were too high. Could the authors update the findings?

9. Line 287, do you mean 0.33% were **not** identified as omicron?

10. Line 336, decreased infectiousness **by age** for children?

Response to Referees' comments

Round 3

REVIEWER COMMENTS

Reviewer #5 (Remarks to the Author):

R3 comments:

R3.Q1 I think there are still issues with the regression equation. The left hand side should have $P[Y_{cp}=1]$, not y_{cp} (we use the observed data to provide information about the (unobserved) probability and in turn estimate the regression coefficients). The main effect of vaccine status of the contact appears to be missing from the regression equation.

AU The reviewer is correct regarding the baseline effect of $VaccineStatus_c$, which did not appear in the estimation equation, but was included in the model, as shown in Tables S17-S20. We apologize for this typo. This is now corrected. (line 131)

Regarding having $\Pr(y_{c,p}=1)$ instead of $y_{c,p}$ on the right hand side of the equation, we are happy to accommodate the reviewers request for changing the notation. (line 131)

R4 comments:

R4.Q1 I think the authors have come up with some creative sensitivity analyses to address potential limitations raised by the reviewer, and I am convinced by them. My only comment would be that these are not particularly well explained in the main text. I understand that space is an issue, but the result is that the reader has to go to the Supplementary to understand these analyses, despite the attempted explanation in the text. For example: “we found a limited effect of 5% of misclassification of secondary cases being infected by the community and not the household” (line 297) – I don’t understand this sentence.

AU Thank you for the positive comment. We have now elaborated on our robustness analyses in the text so they should be easier to understand without having to read the appendix. (lines 151-163 and 299-316)

Other comments:

R4.Q2 Line 231: OR estimates for vaccine status of the primary case are conditional on primary case variant, contrary to what the authors say (variant is included in the model, therefore other ORs in the model are conditional on it). What they mean is that there was no interaction term between vaccine status and variant.

AU The reviewer is correct. Corrected (lines 239-240).

R4.Q3 Line 287: I think a word is missing here: “0.33% of the _____ cases identified by the Variant PCR test were in fact Omicron”?

AU Corrected. (line 296)

Reviewer #6 (Remarks to the Author):

R6 The manuscript by Lyngse et al. compared the transmission dynamics between delta and omicron. The study identified a unique timing where both variants are circulating, and in a household setting where the primary and secondary cases can be more easily identified. The study provided good evidence that the rapid spread of omicron was mainly driven by immunity escape. There is a major issue on the specification of the regression model which may affect the estimates.

Major comments

R6.Q1 1. The study design was well described.

AU We thank the reviewer for this.

R6.Q2 2. Appendix 4.2 nicely considered different scenarios that contacts could potentially be primary cases (Type A & B). Could the authors provide the numbers in each category to support the classification and the likely impact of misclassification?

AU Thanks for the suggestion. This is now included. (appendix Table S14)

R6.Q3 3. For type A & B, was it possible to utilize additional information, such as timing of the symptom onset, testing results of the contacts before confirmation of the first case in the household? This will enhance accuracy of the classification.

AU Yes, we do use timing and test patterns to address this. We restrict our sample to only include households, where no one tested positive within the 60 days preceding the primary case. We do not have data on symptoms or symptom onset. If we had to exploit the testing of contacts in the days preceding the primary case test date, we would need to assume that, e.g., contacts tested negative on day $t-1$ would also test negative on day t . We do not believe we have support for any such assumption. Thus, using the data on tests preceding the primary case will not have any impact on the classifications. We have now elaborated on the robustness analyses in the discussion, to better explain the approach and results.

R6.Q4 4. "We followed all tests of other household members in the study period." Could test results before confirmation of the primary case be used to improve classification of the primary and secondary cases?

AU Please see our answer to R6.Q3 above.

R6.Q5 5. The authors have carried out a series of sensitivity analyses and the results should be robust, though above could be opportunity to reduce misclassification.

AU Thank you for acknowledging the robustness of our results. As also stated in our answer to R6.Q3, we do not believe that using the test results preceding the primary case will change our sensitivity analysis. Furthermore, in these sensitivity analyses, we restrict on an outcome, namely the fact that contacts are obtaining a test and that the test is negative.

R6.Q6 6. Model equation, the interaction term Variant_p x VaccineStatus_c was included. However, it is unclear why the main effect VaccineStatus_c was not included in the model. This is equivalent to making a strong and likely incorrect assumption that vaccination has no protective effect on the contacts within household.

AU Apologies for this typo in the estimation equation. The baseline effect of VaccineStatus_c was included in the model, as also shown in Tables S18-S21. This is now corrected. (line 131)

R6.Q7 7. Could the authors comment on whether Ct values for omicron vs delta have the same interpretation?

AU The interpretation of Ct values of SARS-CoV-2 is independent of the variants of concern, i.e., the same cut-off was applied for the different variants over time. Furthermore, there is no indication of differences in viral load trajectories and peaks between Delta, BA.1 and BA.2, see Non-hospitalised, vaccinated adults with COVID-19 caused by Omicron BA.1 and BA.2 present with changing symptom profiles compared to those with Delta despite similar viral kinetics (<https://www.medrxiv.org/content/10.1101/2022.07.07.22277367v1>).

Minor comments

R6.Q8 8. Introduction, “The Omicron VOC has been reported to be three to six times as infectious as previous variants”. Reference 4 was based on very early estimates in late 2021 and 3-6 times were too high. Could the authors update the findings?

AU We agree that this sentence was based on early estimates, and have now changed the wording of the sentence to reflect that. We agree that newer research (including this manuscript) has produced much lower estimates in the region of +50%, e.g., <https://www.medrxiv.org/content/10.1101/2022.02.15.22271001v1.full.pdf+html>.

R6.Q9 9. Line 287, do you mean 0.33% were *not* identified as omicron?

AU Yes. Corrected.

R6.Q10 10. Line 336, decreased infectiousness *by age* for children?

AU Corrected.